# Fractionation-Dependent Radiosensitization by Molecular Targeting of Nek1

**DOI:** 10.3390/cells9051235

**Published:** 2020-05-16

**Authors:** Isabel Freund, Stephanie Hehlgans, Daniel Martin, Michael Ensminger, Emmanouil Fokas, Claus Rödel, Markus Löbrich, Franz Rödel

**Affiliations:** 1Department of Radiotherapy and Oncology, University Hospital, Goethe University Frankfurt, Theodor-Stern-Kai 7, 60590 Frankfurt am Main, Germany; isabel.freund@tu-darmstadt.de (I.F.); stephanie.hehlgans@kgu.de (S.H.); daniel.martin@kgu.de (D.M.); emmanouil.fokas@kgu.de (E.F.); claus.roedel@kgu.de (C.R.); 2Radiation Biology and DNA Repair, Technical University of Darmstadt, Schnittspahnstrasse 13, 64287 Darmstadt, Germany; ensminger@bio.tu-darmstadt.de (M.E.); lobrich@bio.tu-darmstadt.de (M.L.); 3Frankfurt Cancer Institute (FCI), Theodor-Stern-Kai 7, Goethe University Frankfurt am Main, 60590 Frankfurt am Main, Germany; 4German Cancer Research Center (DKFZ), ImNeuenheimer Feld 280, 69120 Heidelberg, Germany; 5German Cancer Consortium (DKTK) partner site: Frankfurt, 60590 Frankfurt am Main, Germany

**Keywords:** Nek1, cervical cancer, colorectal cancer, fractionation, radiosensitization, xenograft, prognostic marker

## Abstract

NIMA (never-in-mitosis gene A)-related kinase 1 (Nek1) is shown to impact on different cellular pathways such as DNA repair, checkpoint activation, and apoptosis. Its role as a molecular target for radiation sensitization of malignant cells, however, remains elusive. Stably transduced doxycycline (Dox)-inducible Nek1 shRNA HeLa cervix and siRNA-transfected HCT-15 colorectal carcinoma cells were irradiated in vitro and 3D clonogenic radiation survival, residual DNA damage, cell cycle distribution, and apoptosis were analyzed. Nek1 knockdown (KD) sensitized both cell lines to ionizing radiation following a single dose irradiation and more pronounced in combination with a 6 h fractionation (3 × 2 Gy) regime. For preclinical analyses we focused on cervical cancer. Nek1 shRNA HeLa cells were grafted into NOD/SCID/IL-2Rγc−/− (NSG) mice and Nek1 KD was induced by Dox-infused drinking water resulting in a significant cytostatic effect if combined with a 6 h fractionation (3 × 2 Gy) regime. In addition, we correlated Nek1 expression in biopsies of patients with cervical cancer with histopathological parameters and clinical follow-up. Our results indicate that elevated levels of Nek1 were associated with an increased rate of local or distant failure, as well as with impaired cancer-specific and overall survival in univariate analyses and for most endpoints in multivariable analyses. Finally, findings from The Cancer Genome Atlas (TCGA) validation cohort confirmed a significant association of high Nek1 expression with a reduced disease-free survival. In conclusion, we consider Nek1 to represent a novel biomarker and potential therapeutic target for drug development in the context of optimized fractionation intervals.

## 1. Introduction

Genetic screens for cell cycle mutants in the fungus Aspergillus nidulans resulted in the discovery of never-in-mitosis gene A (NIMA), a dual serine/threonine kinase required for mitotic entry [1]. In mammals, 11 homologs named never-in-mitosis kinase-related kinases (Nek1-Nek11) have been reported with a 40% amino-acid similarity in their conserved kinase domains [2,3]. Different lines of evidence indicate a prominent role of Nek1, the prototypic member of the family, in the DNA damage response (DDR), shown by its interaction with proteins involved in DNA repair pathways, cell cycle regulation and apoptosis [4,5]. Nek1 is significantly up-regulated in cells exposed to ionizing radiation, alkylating agents, UV, cross linking agents and oxidative stress [6,7] and colocalizes with markers of the early DDR, such as phosphorylated H2AX [6,8]. Furthermore, a mechanistic role for Nek1 during homologous recombination (HR) was recently reported where it phosphorylates Rad54 at residue Ser572 in a G2 phase specific manner [9]. This promotes Rad51 removal from chromatin and fosters proper completion of HR. Furthermore, a tousled-like kinase 1 (TLK1) > Nek1 > Ataxia Telangiectasia Mutated and Rad3-related (ATR) > Chk1 axis in mediating the DDR and cell cycle checkpoint has been reported [10]. TLK1 phosphorylates Nek1 at residue T141 that in turn is required for Chk1 and Chk2 activation via its association with an ATR and ATR-interacting protein (ATRIP) complex to prime ATR for efficient DNA damage signaling [11]. Consequently, irradiation of fibroblasts with a dominant negative form of Nek1 or transiently depleted Nek1 resulted in spindle defects, abnormal chromosome segregation and prevented G1/S or G2/M phase arrest [12,13]. Furthermore, the TLK1/Nek1 axis mediates a persistent voltage-dependent anion channel 1 phosphorylation and prevents cytochrome C release from mitochondria [14,15] and the execution of apoptosis.

Notably, additional members of the Nek family have been shown to play a pivotal role in the DDR [1,16]. For example Nek4 impacts on entry into replicative senescence and the response to DNA damage by regulating DNA-dependent protein kinase recruitment to DNA double-strand breaks [17], while a complex of Nek5 and topoisomerase I is formed after etoposide treatment in human cancer cell lines [18]. A recent study further indicates that the knockout of Nek8 in murine embryonic fibroblasts results in cellular sensitivity to the replication inhibitor hydroxyurea [19]. In addition, depletion of Nek9 leads to replication stress hypersensitivity, spontaneous accumulation of DNA damage and RPA70 foci formation [20]. Finally, Nek11 prevents cell cycle progression to mitosis [21] and plays a role in the S-phase checkpoint [22].

Only a few data are available on a prognostic or predictive value of Nek1 in cancer. In human glioma tissue Nek1 expression was significantly up-regulated as compared to non-cancerous tissue and correlated with tumor grading and poor survival [23]. Nek1 is part of a 12-gene tumor score to predict progression of non-invasive to muscle-invasive bladder cancer [24] and up-regulation correlated with adverse prognosis in pancreatic cancer [25]. Here, we report that Nek1 knockdown (KD) in combination with a 6 h fractionation regime resulted in radiation sensitization and growth delay in a xenograft model of cervical cancer suggesting that Nek1 might represent a valuable molecular target. We focused on cervical and colorectal carcinomas as a proof of Nek1-mediated radiosensitization as both entities are among the most common cancers diagnosed in humans and cover important causes of mortality worldwide [26]. In addition, we investigated if Nek1 expression levels could represent a prognostic marker for tumor response and analyzed uterine cervical carcinomas. We observed that Nek1 overexpression correlates with an impaired survival in patients treated with chemoradiation and brachytherapy, suggesting that Nek1 represents a radiation resistance marker.

## 2. Materials and Methods

### 2.1. Cell Culture 

Establishment of inducible small-hairpin RNA (shRNA) human cervix carcinoma cell lines (HeLa shCtrl and shNek1) was described in more detail in [9]. Cells were maintained in Dulbecco’s Modified Eagle’s Medium (DMEM, ThermoFisher Scientific, Frankfurt, Germany) supplemented with 10% v/v fetal bovine serum (FBS, ThermoFisher Scientific), 100 units penicillin/mL/100 µg Streptomycin/mL (P/S, Merck, Darmstadt, Germany) and 0.2 µg/mL puromycin (AppliChem, Darmstadt, Germany). Colorectal cancer HCT-15 cells were obtained from the American Type Culture Collection (ATCC, Manassas, VA, USA) and cultured in Roswell Park Memorial Institute (RPMI) medium 1640 (ThermoFisher Scientific, Frankfurt, Germany) (10% FBS, P/S). All cell lines were kept at 37 °C and 5% CO_2_ in a humidified atmosphere.

### 2.2. RNA Interference-Mediated Knockdown 

In vitro shRNA synthesis in HeLa shCtrl (sense, 5′-CTCTCGCTTGGGCGAGAGTAAG-3′) and shNek1 (sense, 5′-GAAATACAGCAATTATTTA-3′) [9] cell lines was induced by incubation with 2 µg/mLdoxycycline (Dox, AppliChem) in cell culture medium changed daily over a period of 5 days. Nek1 KD in HCT-15 cells was accomplished by small interfering RNA (siRNA) transfection using the Roti-Fect PLUS reagent (Carl Roth, Karlsruhe, Germany) protocol with a final concentration of 25 nMNek1-specific (Nek1-1_sense, 5′-GGAGAGAAGUUGCAGUAUU-3′;Nek1-2_sense, 5′-GGG- AAGCUAUGCAGAAUAA-3′) (Eurofins Genomics, Ebersberg, Germany) or unspecific negative control siRNAs (Qiagen, Hilden, Germany). 

### 2.3. D Culture and in vitro Irradiation 

Except for RNA, Caspase 3/7 and protein analyses, cells were plated in culture medium into a laminin-rich extracellular matrix (lrECM, Biozol, Eching, Germany, 0.5 mg/mL) on day 4 of the Dox treatment or 24 h after siRNA transfection. Following an overnight incubation, single doses of 2, 4 or 6 Gy were applied by a 6 MV linear accelerator (Synergy FL, Elekta, Crawley, UK) with a dose rate of 6 Gy/min and a focus-to-isocenter distance of 100 cm. In case of fractionation schedules, fractions of 2 Gy were applied every 2 h, 6 h or 24 h to reach total doses of 6 Gy (3 × 2 Gy). Non-irradiated control cells were exposed to the environmental conditions of the irradiation control room.

### 2.4. Quantitative Nek1 Real-Time Polymerase Chain Reaction (PCR) 

Total RNA was isolated with a NucleoSpin RNA Kit (Macherey-Nagel, Düren, Germany) and 500 ng of RNA were reverse-transcribed by using M-MLV Reverse Transcriptase (Promega, Heidelberg, Germany) and random hexamer primer (ThermoFisher Scientific, Frankfurt, Germany). Subsequently, Nek1 and endogenous reference control (RPL37A) cDNA was quantified in a qPCR using the RT**^2^** SYBR Green qPCR Mastermix (Qiagen, Hilden, Germany) and primer pairs: Nek1_forward, 5′-AGAGGATCAGATTTTGGACT-3′, Nek1_reverse, 5′-GCTCTACAGTACT-ATTAAGAAC-3′; RPL37A_forward, 5′-TGTGGTTCCTGCATGAAGACA-3′; RPL37A_reverse, 5′-GTGACAGCGGAAGTGGTATTGTAC-3′ (Eurofins Genomics, Ebersberg, Germany). PCR cycling conditions were 10 min at 95 °C and 40 cycles switching between 95 °C (15 sec per cycle) and 60 °C (1 min per cycle) in a QuantStudio 5 Real-Time PCR System (ThermoFisher Scientific, Frankfurt, Germany). Fold change in Nek1 expression was calculated by the 2^−∆∆Ct^ method with RPL37A values as reference relative to control cells. 

### 2.5. Immunoblotting

Western immunoblotting was performed as described in [27]. Briefly, cells were lysed in radio-immune precipitation assay buffer supplemented with protease inhibitors. Equal amounts of proteins (35–60 μg) as determined by a micro BCA-protein assay (Pierce, Rockford, IL, USA) were separated on 8% SDS polyacrylamide gels and transferred to a nitrocellulose membrane (Hybond C, Amersham, Freiburg, Germany). Next, membranes were incubated with primary Nek1 (GTX130828, Biozol; Eching) or β-actin antibodies (A5441, Sigma-Aldrich, Munich, Germany) at a 1:1000 or 1:10,000 dilution, respectively. For detection, blots were incubated with goat anti-rabbit (#sc-2054, dilution 1:1000) or goat anti-mouse (#sc-2055, dilution 1:10,000) horseradish peroxidase-linked antibodies (Santa Cruz, Heidelberg, Germany) and visualized with chemiluminescence (Pierce ECL Western Blotting Substrate, ThermoFisher Scientific) and the Odyssey Fc imaging system (LI-COR Biosciences, Bad Homburg, Germany). Individual bands were quantified using the Image Studio software Version 5.2 (LI-COR Biosciences, Bad Homburg, Germany).

### 2.6. 3D Clonogenic Radiation Survival Assay

Clonogenic survival of 3D cultured cells was determined as previously described [27]. In brief, colonies of a size exceeding 50 cells at 7–8 days after plating were considered to be surviving clones and taken into account in the calculations for plating efficiency (formed colonies/plated cells) and for surviving fractions (formed colonies/(plated cells irradiated x plating efficiency non-irradiated cells)).

### 2.7. Cell Cycle Analysis and Apoptosis Assays 

Flow cytometric analysis of cell cycle distribution was performed with a CytoFlex S cytometer (Beckman Coulter, Krefeld, Germany). Briefly, cell-lrECM mixtures were harvested in phosphate-buffered saline (PBS) and single cell suspensions were collected by trypsinization, washed with PBS and fixed with ice cold 80% ethanol for 10 min. After centrifugation (200× g for 5 min), cell pellets were resuspended in PBS containing 40 mg/mL propidium iodide (Merck, Darmstadt, Germany) and 40 mg/mL RNase A (Qiagen, Hilden, Germany) and subjected to analysis. Finally, cells were analyzed using the CytExpert Software (Beckman Coulter). For quantification of Caspase 3/7 activity, a CASPASE GLO™-assay (Promega, Mannheim, Germany) was used according to the manufacturer’s recommendation.

### 2.8. Staining and Quantification of γH2AX Foci Formation 

Following fixation in 3.7% formaldehyde (Carl Roth) and permeabilization with 0.25% triton X-100 in PBS for 15 min at RT, single cells were blocked with 5% BSA for 1 h at room temperature (RT). Next, cells were incubated with a primary mouse anti-phospho-H2AX antibody (Ser139, clone JBW301, #05-636, Millipore, Schwalbach, Germany) for 2 h at RT and with a secondary goat anti-mouse antibody coupled to AlexaFluor 488 (Invitrogen, Darmstadt, Germany) for 1 h. DNA was counterstained with a 1000 ng/mL 4′,6-diamidino-2-phenylindole (DAPI) solution for 5 min at RT and cells were then mixed with mounting medium (Vectashield, Vector Laboratories, Peterborough, UK) and transferred to microscopic slides. Images were taken using an AxioImagerZ1 microscope and Axiovision 4.6. software (Zeiss, Göttingen, Germany) and **γ**H2AX foci formation was quantified for each data point from 50 nuclei and three independent experiments.

### 2.9. Murine Xenograft Model and in vivo Irradiation 

In vivo experiments were approved by the government committee (Regierungspräsidium Darmstadt, Darmstadt, Germany, FK/1098) and were conducted in accordance with the requirements of the German Animal Welfare Act. Female 12- to 16-week-old NOD/SCID/IL-2Rγc−/− (NSG) mice were injected subcutaneously with 1 × 10^6^ HeLa shNek1 and shCtrl cells. After visual detection of tumor nodes, Dox was provided (2 µg/mL + 2% sucrose) in drinking water. At day 10 of the treatment, mice were irradiated by image-guided-radiotherapy (IGRT) using a Small Animal Radiation Research Platform (SARRP, Xstrahl Ltd., Camberley, UK). Animals were subjected to a Cone-Beam CT (CB-CT) operating at 65 kV, 0.5 mA and irradiated while anesthetized with 2.5% isoflurane (AbbVie, Wiesbaden, Germany). Fractionated single doses of 2 Gy every 6 h were applied using a 10 mm collimated beam operating at 175 kV, 15 mA. Tumor growth was monitored by caliper measurements and tumor volumes were calculated using the formula volume = (width^2^ × length)/2 [28]. Finally, tumors were harvested for qPCR and Western blotting or embedded in paraffin for histochemical evaluation.

### 2.10. Patient Characteristics

74 patients with uterine cervix squamous cell carcinoma treated at the University Hospital Frankfurt am Main from 1999 to 2017 were enrolled in our study. Eligibility criteria covered histological proof of cervix carcinoma (Fédération Internationale de Gynécologie et d’Obstétrique (FIGO)) stages Ib to IVb [20] and curative intended chemoradiotherapy (CRT) and brachytherapy (BT). Patients were subjected to pretreatment staging including computer tomography or magnetic resonance tomography of the pelvis and abdomen, chest radiography, and baseline laboratory examination. Written consent and approval from the institutional review board and ethics committee (No. 31/15, University Hospital Frankfurt am Main) was obtained in accordance with the Helsinki Declaration of 1975.

### 2.11. Treatment and Follow-Up

Patients were treated using a linear accelerator (Elekta, Crowley, UK) followed by intracavitary +/- interstitial high dose range (HDR)-BT. External beam radiotherapy (RT) was administered using either a four-field technique (n = 22), intensity modulated RT (IMRT) or 3D conformal RT (n = 52) with a median dose of 50.4 Gy (range, 45.0–66.6 Gy) in 5 weekly fractions of 1.8 Gy. Median total physical BT dose was 40 Gy (range, 4.0–48.0 Gy). Cisplatin-based chemotherapy (20–40 mg/m^2^) was administered weekly or in the first and last week of treatment. In addition, 12 patients received two cycles of 5-Fluorouracil (600 mg/m^2^) and two patients Mitomycin-C (7 mg/m^2^) or Paclitaxel (25 mg/m^2^). Follow-up examinations were scheduled every three months in the first two years, followed by 6-month intervals afterwards. 

### 2.12. Immunohistochemical Staining of Nek1 and Scoring 

Formalin-fixed paraffin embedded (FFPE) biopsies of patients or xenograft tumor specimens were subjected to a standardized procedure with a DAKO EnVision^TM^ FLEX Peroxidase Blocking reagent (DAKO, Hamburg, Germany). Primary anti-Nek1 antibodies (HPA040413, Merck, Darmstadt, Germany) were applied for 2 h at RT followed by incubation for 60 min with anti-rabbit secondary antibodies (SM802, DAKO EnVision™ FLEX kit K8000, DAKO). Epitope-antibody conjugates were visualized using dextran polymer conjugated horseradish peroxidase and 3,3′-diaminobenzidine chromogen and hematoxylin solution (Gill 3, Sigma-Aldrich, Munich, Germany) for counterstaining. Negative controls were included by staining in the absence of the corresponding primary antibody. Nek1 immunoreactivity was assessed considering the fraction of positive tumor cells (1: (0–25%), 2: (26–50%), 3: (51–75%) and 4: (>75%)) and the intensity of staining scored as 1 (weak), 2 (moderate) and 3 (intense). These parameters were multiplied to produce an individual weighted score (WS). We defined a WS of >6 as “high” and a WS ≤6 as “low” expression. Samples were evaluated by two investigators without knowledge of the clinicopathologic or clinical data to minimize interobserver variability and image acquisition was via AxioImagerZ1 micro- scope and Axiovision 4.6 software (Zeiss, Göttingen, Germany). Quantification of histochemical p16^INK4a^ detection has been previously reported in detail in [29].

### 2.13. Cervical Cancer TCGA Dataset

The Cancer Genome Atlas (TCGA) RNA-Sequencing (RNA-Seq) and associated clinical data for the cervical cancer cohort were retrieved from cBioPortal (https://www.cbioportal.org/). We included only the patients treated with CRT in our analysis, resulting in several n = 90 patients with Nek1 expression levels and clinical data available regarding disease-free survival (DFS). The cut-off for Nek1 expression levels (185.44 counts) was calculated using the maximally selected rank statistics that calculates the most optimal cut-off for continuous variables using log-rank statistics. This analysis was done using the R package “survminer” (Version 0.4.6, R foundation, Vienna, Austria).

### 2.14. Statistical Evaluation

Experimental data are presented as mean +/− standard deviations from at least three or more independent experiments. Levels of significance were calculated using the Student’s T-test for independent samples (EXCEL 2010, Microsoft, Munich, Germany). The cumulative incidence of local failure was defined as the time from start of CRT to the first local tumor detection (i.e., any local tumor recurrence after initial complete response). The cumulative incidence of distant failure was defined as the time from start of CRT to any occurrence of distant metastasis during CRT or during follow-up. Data from patients who were alive and free of recurrences or died without having a recurrence were censored for these endpoints. Overall (OS) and cancer-specific survival (CSS) were defined as the time of start of CRT to death for any reasons or to cancer-related death, or the day of the last follow-up. Survival was plotted according to the Kaplan-Meier method. Log-rank testing and Cox proportional hazard modeling were used for univariate and multivariate analyses using IBM SPSS Version 25 Software (IBM, Ehningen, Germany). For all statistical analyses a *p* < 0.05 was considered statistically significant.

## 3. Results

### 3.1. Knockdown of Nek1 Reduces 3D Clonogenic Cell Survival 

Recent analyses indicate an involvement of Nek1 in the regulation of DNA damage repair by HR as shown for fibroblasts and HeLa cervical tumor cells [9]. Here, we used HeLa and HCT-15 cells from a colorectal adenocarcinoma [30]. Nek1 KD HeLa cells carrying an inducible shRNA against Nek1 were generated by lentiviral transduction; Nek1 KD in HCT-15 cells was achieved by transient siRNA transfection. As depicted in Figure 1A, following incubation with Dox for 5 days or transfection with siRNA for 48 h, a significant (*p* < 0.001) decrease in Nek1 mRNA and protein levels was evident. Densitometric evaluations provided KD efficiencies of 80% for HeLa and 70% for HCT-15 cells. Using these Nek1 KD cells, we observed significantly (*p* < 0.05) diminished colony formation abilities after single dose X-irradiation in 3D survival assays (Figure 1B,C), consistent with earlier findings that a depletion of Nek1 confers sensitivity to genotoxic stress including irradiation [6,8,9].

### 3.2. Fractionation-Dependent Radiation Sensitization by Knockdown of Nek1

As Nek1 is reported to impact on the HR repair pathway [9], a Nek1 KD is expected to impact on DNA repair most pronounced in the G2 phase. Accordingly, we next assessed cell cycle distributions after irradiation of Nek1 KD HeLa and HCT-15 cells and used fractionated irradiation to increase the number of cells in G2 (Figure 2A). As depicted in Figure 2B,C, a 3 × 2 Gy irradiation and 6 h fractionation regime resulted in a significantly (*p* < 0.01) increased proportion of HeLa and HCT-15 Nek1 KD cells in G2 compared to a 2 h or 24 h fractionation interval. The analysis of HeLa shCtrl cells in the presence of Dox, shNek1 cells in the absence of Dox and HCT-15 siCtrl transfected cells indicates that this enrichment was not dependent on Nek1 attenuation (Appendix A). 

We next performed 3D clonogenic assays comparing the 2, 6, and 24 h fractionation intervals. As shown in Figure 2D,E, single dose irradiations and 3 × 2 Gy irradiations with a 2 h interval caused similar survival curves for both Nek1 KD and control cells. The 24 h fractionation format resulted in increased clonogenic survival compared to single dose irradiations for both Nek1 KD and control cells and did not enhance the radiosensitizing effect of the Nek1 KD. Importantly, for the 6 h fractionation format, we observed increased radiation resistance in control cells but decreased resistance in Nek1 KD cells compared to the single dose irradiations. As a result of this, the radiosensitizing effect of the Nek1 KD was substantially greater for the 6 h scheme compared to single dose irradiations or the 2 and 24 h fractionation schemes. This sensitizing effect was not observed in HeLa shNek1 cells irradiated in the absence of Dox (Appendix A).

Radiation sensitization by Nek1 KD is considered multifactorial involving multiple pathways. Thus, we next asked whether a Nek1 KD in the 6 h fractionated regime impacts on the DDR and apoptosis induction at 24 h after treatment. We observed a slight but significant (*p* < 0.01) increase in the number of residual (24 h) *γ*H2AX foci in Nek1 KD compared with control cells (Figure 3A,B) together with a slightly increased fraction of apoptotic cell death assayed by a Caspase 3/7 activity assay (Figure 3C).

The data depicted so far indicate a radiation sensitizing effect of Nek1 KD most pronounced if combined with a 6 h fractionation interval. To independently confirm a Nek1 KD radiation sensitizing effect in a 6 h fractionation schedule, we next focused on cervical carcinomas for the following in vivo experiments and the analysis of clinical samples. A xenograft model was established by subcutaneous injection of HeLa shNek1 and shCtrl cells into NSG mice. After the tumors reached an average size of 190 mm^3^, a significant (*p* < 0.001) Dox-dependent depletion of Nek1 was confirmed on the level of mRNA and by histochemical detection (Figure 4A,C). Furthermore, a densitometric assessment of protein expression levels in tumor tissues revealed a KD efficiency of 80% (Figure 4A). Next, mice were treated with 3 × 2 Gy fractions in a 6 h interval to reach a total dose of 6 Gy and tumor growth was monitored by analyzing tumor volumes. As illustrated in Figure 4B,D, fractionated irradiation of control tumors only marginally reduces tumor growth. In contrast, fractionated irradiation of Nek1 KD tumors caused a significant (*p* < 0.001) reduction of tumor volume within the first days after irradiation and a subsequent growth retardation compared to non-irradiated animals. This sensitization was not observed in HeLa shCtrl tumors either in the presence or absence of Dox (Appendix A).

### 3.3. Nek1 Overexpression in Cervical Cancer is Associated with Impaired Clinical Outcome

The in vitro and in vivo findings reported so far indicate a radiation sensitization upon KD of Nek1. We next investigated Nek1 expression in tumors of patients treated with CRT and focused our analyses on pretherapeutic samples from patients with cervical cancer treated at a single university center.

We identified 28 patients (37.8%) with increased Nek1 expression (weighted score: WS > 6) and 46 patients (62.2%) with low Nek1 expression in tumor biopsy samples (WS ≤ 6; Figure 5A). Individual scores (scatter plots) are given in Figure 5B. We did not detect a significant relationship of Nek1 expression with age, TNM and FIGO category, grading and p16Ink4a expression (Appendix A). In univariate analyses, local tumor failure was associated with T-stage (*p* = 0.011), FIGO category (*p* = 0.006) and Nek1 overexpression (*p* = 0.028) (Table 1, Figure 5C), while only Nek1 overexpression remained an independent adverse prognosticator for local failure in multivariate analyses (Table 1). High Nek1 expression (*p* = 0.035; Figure 5D), T-stage (*p* = 0.011) and FIGO category (*p* = 0.008) were associated with distant metastasis risk; only Nek1 (*p* = 0.025) continued to predict the distant metastasis risk in multivariate analyses (Table 1). Clinical factors with a significant adverse impact on CSS were advanced T-category (*p* = 0.006), FIGO category (*p* = 0.017) and p16INK4a detection (*p* = 0.013).

As depicted in Figure 5E, Nek1 overexpression was further significantly associated with an impaired CSS (*p* = 0.008) in univariate analyses, and in line with p16INK4a expression (*p* = 0.023) remained a significant independent predictor for CSS (*p* = 0.001) in multivariate assessments (Table 1). In addition, high Nek1 expression was associated with an impaired OS (*p* = 0.041, Figure 5F) in univariate analyses but lost its significance in multivariate testings. To validate the findings regarding the prognostic role of Nek1, we finally analyzed data from a cervical cancer patient cohort derived from the TCGA databank. From the data available, with a cut-off level of 185.44 counts we discovered a significant (*p* = 0.020) association of a high Nek1 gene expression and poor DFS (Figure 5G) confirming our immunohistochemical findings.

## 4. Discussion

The mechanisms underlying the association of Nek1 with the radiation response in cancer cells and the response to radiation therapy remain to be defined. They may, however, be related to the interrelationship of the kinase with mechanism(s) of proliferation/cell cycle regulation, DNA repair, and resistance to cell death as Nek1 is involved in every one of these processes and puts them to work in a coordinated manner [1].

The importance of Nek1 in cell cycle regulation has been characterized most thoroughly in response to ionizing irradiation (IR). Nek1 has been reported to be involved in the induction of an arrest at the G1/S and G2/M transition, and Nek1 KD cells fail to activate the checkpoint kinases Chk1 and Chk2 in response to UV, IR, and H_2_O_2_ treatment [4,5,8]. Notably, these studies used non-transformed cells established from kat2J/Nek1-/- mice, HK2 human proximal renal tubular epithelial cells and fibroblasts [4,5] that may not reflect the situation in malignant cells. Indeed, in the present study, we observed that the attenuation of Nek1 in HeLa and HCT-15 carcinoma cells does not interfere with the ability to induce a G2/M arrest.

Our group has recently established that Nek1 is implicated in HR repair with a G2-phase specific phosphorylation of Rad54 at residue Ser572 to promote Rad51 removal from chromatin [9]. As DNA repair by HR is restricted to the late S and G2 phases when a sister chromatid is available as a template [31], we investigated whether the enrichment of cells in the G2 phase by fractionated irradiation may increase the therapeutic effect of a Nek1 attenuation. Indeed, by using a 6 h fractionation schedule, we uncovered a significantly increased radiation response in cells with a Nek1 KD compared to the single dose-treated controls (Figure 2D,E). 

Historically, fractionation regimes were established based on the findings of different radiobiological responses of cancer and normal tissues. Modern schedules involve the standard fractionation regime with daily doses of 1.8–2 Gy given five days a week and aim to maximize tumor cell killing while avoiding normal tissue toxicities [32]. However, due to the lower single doses and the time interval between the fractions that allows DNA damage repair, fractionated regimes are considered to be less effective and to require elevated doses to reach an equal impact, i.e., isoeffect, on cellular survival compared to higher single doses [33]. Although a variety of regimes including hyperfractionation (smaller doses of 0.5–1.8 Gy with multiple fractions per day) and hypofractionation (daily fractions of 3–20 Gy) have been developed in the clinical setting [32,33], it is unclear how they affect molecular targeting therapies such as the attenuation of Nek1. 

Our finding that a 6 h fractionation schedule increases the therapeutic effect of a targeted approach may have several clinical implications. First, a 6 h regime could easily be translated to the clinical practice. In line with this, a meta-analysis of n = 6515 patients with head and neck cancer revealed that two to three fractions delivered each day with a 6 h interval (hyperfractionation) is associated with a significant survival benefit compared to a standard 24 h fractionation [34]. Our finding suggests that this benefit might be further increased when combined with a targeted therapy against Nek1. Second, in line with preclinical evidence that fractionated radiation can induce an extensive array of targetable molecules [35], Nek1 inhibition may be combined with additional targeting strategies against transcription factors, immune and inflammatory responses and tumor cell stemness [36]. Finally, enrichment of cells in G2 by fractionated irradiation may represent a more general approach to increase the therapeutic efficacy of HR-specific therapeutics such as small molecule inhibitors of Rad51 [37,38].

We also assessed Nek1 expression in patients treated with CRT and focused on histochemical analyses of patients with cervical cancer treated at a single center. For all stages of the disease, the 3-year survival rate is less than 50% in non-developed countries [39] and 68% in developed countries [40], but is stagnating in recent years. Thus, more effective treatment strategies are required, including the need to develop biomarkers for patient stratification and novel targeted therapies [41]. Notably, only a limited number of such markers is currently available to predict therapy response and prognosis in this tumor entity. These markers include clinical parameters such as the tumor size, FIGO stage, parametrial and lymph node invasion and the availability of BT [42] but also molecular markers such as the expression of the epidermal growth factor receptor [43].

To the best of our knowledge, our study is the first to report that a high expression level of Nek1 is associated with an impaired clinical outcome following CRT and BT in patients with cervical cancer. In detail, Nek1 significantly impacts on local and distant recurrence, CSS and OS (Figure 5) and the association with Nek1 remains significant for most endpoints in multivariate assessments (Table 1). These findings are in line with the few reports available in the literature on a prognostic value of Nek1 expression in different cancer entities. Up-regulation of Nek1 in glioma tissue correlated with the clinical grade and Karnofsky performance scale, and was significantly associated with a poor OS, while an siRNA-mediated KD of Nek1 sensitized glioblastoma cells to the drug temozolomide [23]. Nek1 expression was elevated in thyroid tumors with multifocality and in patients with lymph node metastasis [44]. Nek1 is further reported to be associated with a decreased sensitivity to DNA damaging agents in renal cell carcinomas [45] and to be part of a 12-gene tumor signature to predict progression of non-invasive to muscle-invasive bladder cancer [24]. 

We also would like to acknowledge some limitations of our work including the retrospective design and the limited pool of patients which could result in a selection and calculation bias. While confirmation of our findings in a larger cohort of patients and additional tumor entities including rectal carcinomas is indicated, we aimed to address this limitation with the analysis of TCGA data for cervical cancer as an independent validation cohort. Again, we discovered a significant correlation of Nek1 expression with disease outcome, confirming the association of this protein with prognosis and treatment response and further highlighting the role of Nek1 as a therapy resistance factor.

## 5. Conclusions

Our results about the association of Nek1 with the clinical outcome, its impact on clonogenic survival and tumor growth in vivo indicate the protein to represent a valuable target for radiation sensitization. In addition, given an overexpression of multiple members of the Nek family in human malignancies and overlapping functional properties [16,44], the entire family should be considered for use in pharmacological approaches to increase radiation or chemotherapy response in anti-cancer treatment.

## Figures and Tables

**Figure 1 cells-09-01235-f001:**
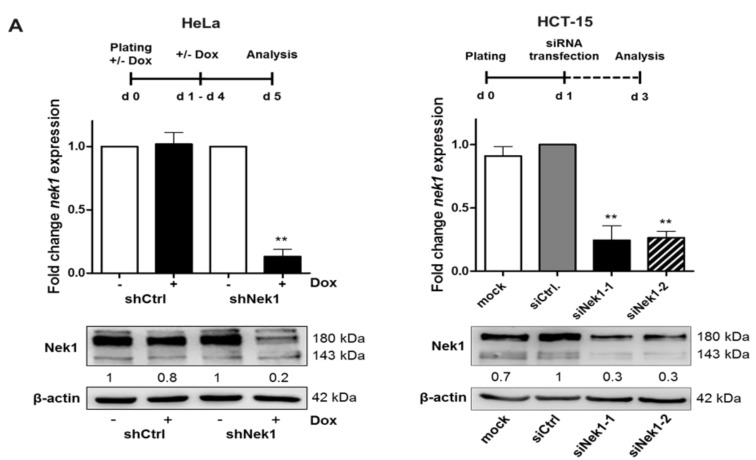
(**A**) HeLa shNek1 cells were incubated for a period of five days with 2 µg/mL doxycycline (Dox) and HCT-15 cells were treated for 48 h with Nek1 specific siRNA (25 nM). Stable non-specific shCtrl expressing HeLa cells and mock (Roti-Fect) or non-specific siCtrl-treated HCT-15 cells served as a control. Shown are the relative mRNA levels of Nek1 in reference to RPL37A expression normalized to HeLa shCtrl and HCT-15 siCtrl cells. Representative Western blots from at least three independent experiments are shown. Numbers indicate protein expression relative to β-actin and normalized to shCtrl–Dox, shNek1–Dox, or siCtrl. (**B**) HeLa or HCT-15 cells were plated in culture medium into a laminin-rich extracellular matrix on day 4 of the Dox treatment or at 24 h after siRNA transfection and irradiated 24 h later. Radiation survival following 2, 4, or 6 Gy single dose irradiation was analyzed by 3D colony forming assays. Stable shCtrl expressing HeLa cells and mock- or siCtrl-treated HCT-15 cells served as controls (for all graphs means ± SD; n ≥ 3; * *p* < 0.05, ** *p* < 0.01 vs. control). (**C**) Representative image of 3D-grown colonies of HeLa shNek1 cells in the presence or absence of Dox (left) and HCT-15 cells transfected with siNek1-2 or siCtrl (right) following a 4 Gy exposure. Bars correspond to 100 μm.

**Figure 2 cells-09-01235-f002:**
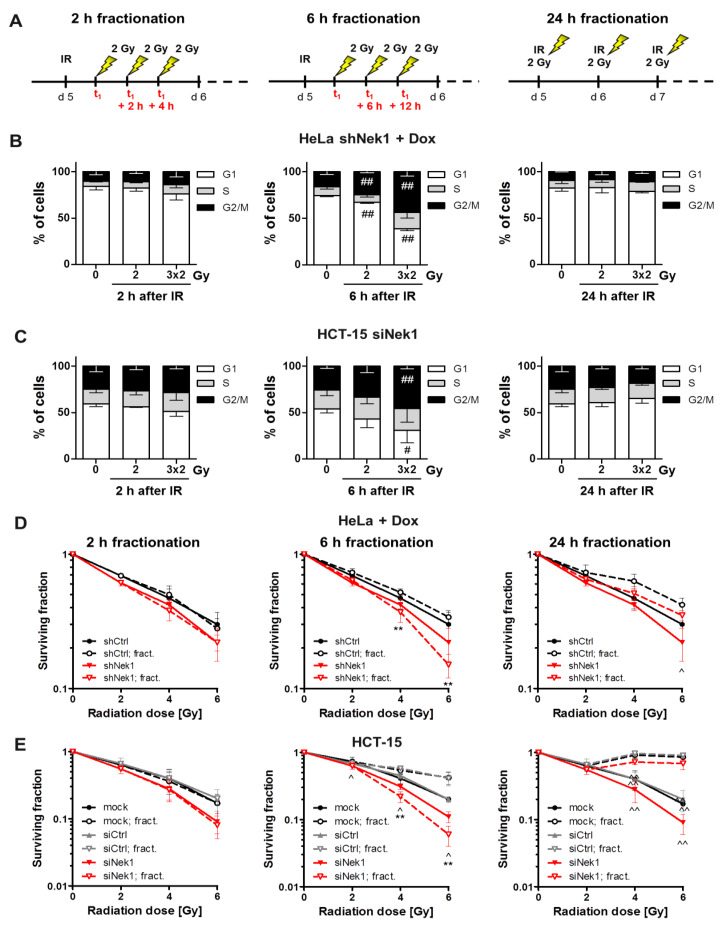
(**A**) Schematic representation of the different fractionation schedules. Cell cycle distribution of Nek1 KD HeLa (**B**) and HCT-15 cells (**C**) at 2 h. 6 h and 24 h after irradiation with either a single dose of 2 Gy, or fractionated 3 × 2 Gy with an interval of 2 h (left), 6 h (middle) and 24 h (right) analyzed by flow cytometry (n = 3; # *p* < 0.05, ## *p* < 0.01 irradiated vs. non-irradiated cells). 3D clonogenic radiation survival following a 2, 4 or 6 Gy single dose irradiation or fractionated irradiation with 2 h (left), 6 h (middle) and 24 h interval (right) of Nek1 KD HeLa (**D**) and HCT-15 cells (**E**). Stable shCtrl expressing HeLa cells and mock (m) or control siRNA-treated HCT-15 cells, treated in a similar manner, served as a control (for all graphs means ± SD; n = 3; * *p* < 0.05, ** *p* < 0.01 Nek1 KD vs. control, ^ *p* < 0.05, ^^ *p* < 0.01 fractionated vs. single dose).

**Figure 3 cells-09-01235-f003:**
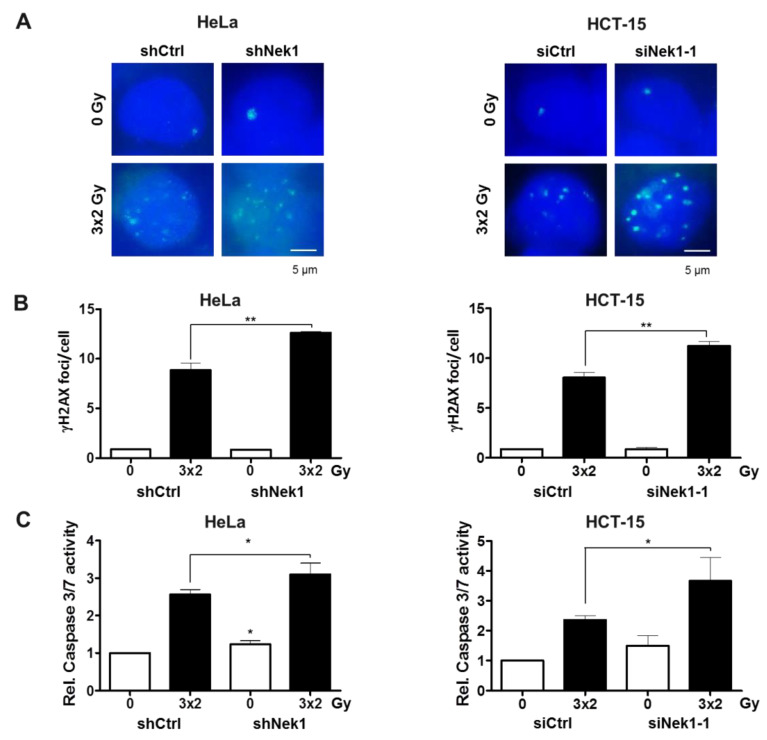
(**A**) Representative images of *γ*H2AX foci from non-irradiated and 3 × 2 Gy-irradiated control and Nek1 KD Hela and HCT-15 cells are shown. Nuclei were counterstained with DAPI. Scale bar, 5 μm. (**B**) Residual *γ*H2AX foci and (**C**) apoptosis induction (24 h after the last fraction) of Dox-treated HeLa shCtrl and shNek1 (left) and HCT-15 cells (right) transfected with non-specific Ctrl and Nek1 siRNAs following a 3 × 2 Gy and 6 h interval fractionated irradiation (means ± SD; n = 3; * *p* < 0.05, ** *p* < 0.01 Nek1 KD vs. control).

**Figure 4 cells-09-01235-f004:**
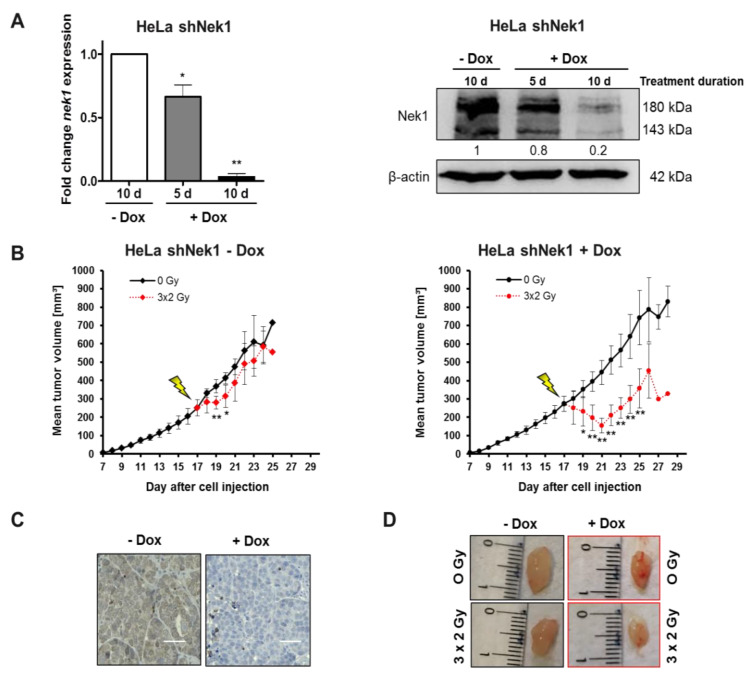
Female, 12- to 16-week-old NOD/SCID (NSG) mice were injected subcutaneously with 1 × 10^6^ HeLa shNek1 and HeLa shCtrl cells in 100 μL PBS. After visual detection of tumor nodes about 10 days post cell injection, Dox was given (2 µg/mL doxycycline hyclate + 2% sucrose) in drinking water for an additional 10 days. Mice were irradiated by image-guided-radiotherapy (IGRT) with three fractionated single doses of 2 Gy every 6 h to reach a total dose of 6 Gy. (**A**) Tumor Nek1 mRNA and protein expression was analyzed by quantitative PCR (left) and Western blotting (right) after a 5 days or 10 days treatment with Dox. Numbers indicate protein expression relative to *β*-actin control and normalized to 10 d–Dox. (**B**) Relative tumor growth curves as monitored by using calipers (6 animals per group) inoculated with HeLa shNek1 cells treated with (right) or without Dox (left). Mean values of tumor volumes for each treatment group are shown (* *p* < 0.05, ** *p* < 0.01, irradiated vs. non-irradiated mice). (**C**) Representative images of histological detection of Nek1 levels in tumor tissue (bars correspond to 50 μm) and (**D**) representative images of tumors after fractionated irradiation with 3 × 2 Gy in Nek1-depleted (+Dox) and non-depleted tumors (-Dox) at day 28 after cell injection.

**Figure 5 cells-09-01235-f005:**
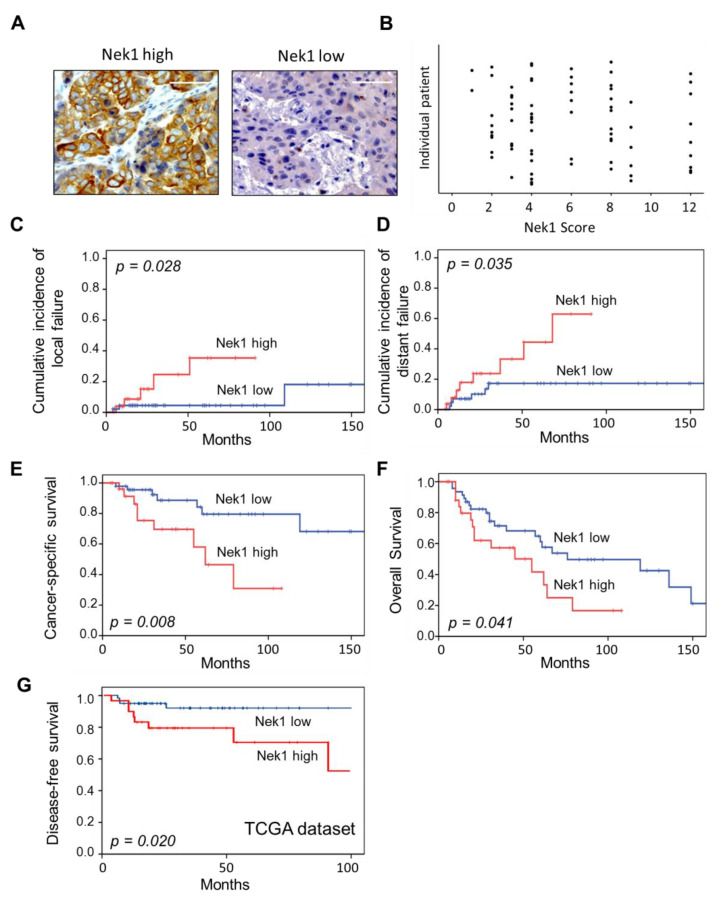
(**A**) Representative examples of uterine cervical cancer with a high (WS > 6) and low (WS ≤ 6) histochemical detection of Nek1 in tumor cells. Original magnification x 400, scale bars correspond to 50 **μ**m. (**B**) Scatter Plot of individual Nek1 weighted scores defined as fraction of positive tumor cells (1: (0–25%), 2: (26%–50%), 3: (51%–75%) and 4: (> 75%)) multiplied by intensity of staining scored as 1 (weak), 2 (moderate) and 3 (intense). Cumulative incidence of local (**C**) and distant failure (**D**), cancer-specific (**E**) and overall survival (**F**) in 74 patients with cervical cancer according to Nek1 expression based on immunohistochemical evaluation of biopsy specimens. (**G**) Disease-free survival in 90 patients with cervix carcinoma derived from the TCGA dataset (https://www.cbioportal.org).

**Table 1 cells-09-01235-t001:** Univariate and multivariate analyses of prognostic markers in patients with cervical cancer treated with chemoradiotherapy (CRT) and brachytherapy (BT). Abbreviation: FIGO: Fédération.

			Multivariate Analyses		
			95% Confidence Interval		
	Univariate *P*-Value	Hazard Ratio (HR)	Lower	Upper	*P*-Value
**Cumulative incidence of local failure**					
T-stage (T1-2/T3-4)	0.011	1.07	0.2	5.7	0.935
FIGO (Ia-IIb/IIIa-IVb)	0.006	5.6	0.5	62.52	0.162
Nek1 (WS ≤ 6/> 6)	0.028	5.46	1.02	29.26	0.047
**Cumulative incidence of distant failure**					
T-stage (T1-2/T3-4)	0.011	2.13	0.45	10.01	0.336
FIGO (Ia-IIb/IIIa-IVb)	0.008	5.59	0.5	62.7	0.162
Nek1 (WS ≤ 6/> 6)	0.035	4.49	1.17	10.41	0.025
**Cancer-specific survival**					
T-stage (T1-2/T3-4)	0.006	5.58	0.69	44.67	0.105
FIGO (Ia-IIB/IIIa-IVb)	0.017	1.01	0.89	11.58	0.989
p16^INK4a^ (WS ≤ 6/> 6)	0.013	3.44	1.18	10.04	0.023
Nek1 (WS ≤ 6/> 6)	0.008	6.31	2.09	10.01	0.001

Internationale de Gynécologie et d’Obstétrique. Significant results have been marked in bold.

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
