# Peer review of "Fractionation-Dependent Radiosensitization by Molecular Targeting of Nek1"

_cells, 2020, doi:10.3390/cells9051235_

Round 1
Reviewer 1 Report
This is a very nice contribution by Freund and collaborators, which shows in a convincing way, that the human protein kinase Nek1 can serve both as a biomarker and even as a potential therapeutic target in cervical cancer. In in vitro Nek1 knock down experiments show that these cells are surviving less to X-ray exposure than control cells. Next, it was observed that cells with less Nek1 survive even less to a 6h three fold fractionation exposure to X-ray and have higher DNA damage and apoptosis levels than control cells. The Nek1 depleted cells when implanted into mice produced smaller tumors compared to control cells , after fractionated X-ray exposure. Finally, data from 74 cervical cancer patients were analyzed. The patients with higher Nek1 expression showed worse survival and Nek1 over-expression emerges as a new and powerfull adverse prognosticator.
The experimentation is very solid and the data are well presented and convincing. This certainly is an important contribution, which suggests that Nek1 is both a marker protein and a possible therapeutic target in cancer, especially as a chemo- or radio-sensitizing target, in cervical cancer and likely in other settings, too. The data also make much sense in comparison with other papers previously published on the molecular functions of Nek1 in the context of the DNA damage response and Homology DNA repair mechanisms.
I think the paper can be basically accepted as presented. My only major suggestion would be the following:
Both the Introduction and discussion focus almost exclusively on Nek1 and its role in DDR and HR. However, the Nek family has ten other members and several studies suggest an involvement of other members (Nek4,5,8,10 and 11) in different aspects of the DNA damage response, DNA repair mechanisms or replication stress. A brief paragraph in the introduction should be added to state this. In the discussion it maybe discussed briefly, too, if Neks in general could be targeted (for chemo- or radio-sensitization) in cancer. This may be compared to the Nek1 example and the also the overall potential of this family as novel targets in cancer therapy.
Author Response
We would like to thank the reviewer for his/her kind evaluation of our manuscript and the insightful suggestion. In a revised version of the manuscript we now include a short paragraph on the involvement of other members of the Nek family in different aspects of the DNA damage response and replication stress (page: 2, lines 65-73) and on Nek family members as molecular targets in cancer therapy (page: 14, lines 467-470).
Reviewer 2 Report
In this article, the authors report the potential role of NIMA (never-in-mitosis gene A)-related kinase 1 (Nek1) as a biomarker and therapeutic target for radiation sensitization in tumor cells. This is an interesting report supported by some solid experiments using RNA interference in in vitro and in vivo approach, and more important clinical data, indicating the importance of Nek1 in cervical cancer. In my opinion is a well performed manuscript. I have some minor comments:
- In the introduction there is not a context of the type of cancer that the authors study in this work. Which is the relation of study cervical and colorectal cancer? Perhaps would be better focus in cervical cancer using another cell line of cervical cancer.
- In material and methods the authors should detail better some methods such as the inducible small-hairpin RNA (shRNA) or the Immunoblotting sections for example.
- The RNA interference sequence used in the inducible shRNA ist the same sequence reported for the siRNA? Which is the sequence for the shControl?
- In material and methods section Immunoblotting the author’s at least should indicate the dilution of antibodies used.
- Figure 2D, E and 3A need representative image, could be in the supplementary information.
- The analysis of the figure 4B should be show in the same graph. Were performed the in vivo experiments of the four groups together? Is the difference showed between the 0 Gy+Dox and the 3x2 Gy+Dox also with the groups –Dox? Does Doxycycline have any effect on the tumor growth?
- Figure 4D should show an image with all the tumors used for the analysis and with all the groups –Dox and +Dox.
- In Figure 5A should add a scatter plot with the score for each patient.
- In the TCGA data please indicate the cut-off and the values used to calculate.
Author Response
- Thank you for this helpful comment. In a revised version of the introduction section we now briefly stated the relation of our study to cervical and rectal carcinoma (page 2, lines 81-83).
- We would like to thank the reviewer for his/her helpful objection. We now describe in more details the methods of RNA-interference and western blotting. For details see answers to point 3 and point 4.
- Due to cloning reasons Nek1 sequences in the shRNA plasmids and sequences of the synthetic siRNA oligonucleotides are not identical, but all sequences used in our investigations were confirmed by blast analyses. Corresponding sequences of shNek1 and shControl are now given in detail in a revised version of the paragraph “RNA interference-mediated knockdown”.
- We apologize for the short description of the method in the primary version of our manuscript and describe now in more detail the method of immunoblotting including dilution of antibodies, detection and analyses.
- In a revised version of Figure 3 we now include representative images of gH2AX foci from both, Hela and HCT-15 cells. Colony forming assays, however, were exclusively performed in a 3D laminin-rich extracellular matrix and accordingly (as compared to standard 2D assays with stained cells fixed on the plastic surface), plates could not be stored. Due the multiple conditions in our fractionated experiments, and the restricted time for revisions, we were not able to acquire time consuming images from every condition and apologize not to provide images from the fractioned experiments.
- In vitro experiments were performed at the same time but differ from their setup with mice in the + Dox group to be treated with the drug in drinking water for the whole duration of the experiment, while – Dox mice were treated with vehicle in another cage. Accordingly, data arise from different groups and due to reasons of clearness we decide to display analysis in two different graphs. Moreover, as depicted in supplemental Figure S2B, an effect of 3 x 2 Gy irradiation was also observed in Nek1 shControl cell inoculated mice in the absence of Dox treatment. Doxycycline treatment did not result in a significant effect on tumor growth in our in vivo experiments.
- In a revised version of figure 4D we now include exemplary images of tumors of all conditions.
- Again, thank you for his/her helpful objection. In a revised version of Figure 5A we now include a scatter plot of all individual scores and have modified the text according to the reviewer´s suggestion.
- In a modified version of the paragraph “Cervical Cancer TCGA dataset” and in the text we now indicate the cut-off level for Nek1 expression (185.44 counts) and more clearly describe the method for calculation.
Reviewer 3 Report
The study by Freund et al assesses the potential of targeting NEK1 in human cancers to improve the effectiveness of radiotherapy regimes (in particular fractionation). To accomplish this, they adopt a stable tet-inducible cell line system for both in vitro and in vivo studies. They find that 6hr 3x2Gy fractionation doses leads to increased cell killing following depletion of NEK1. Although the mechanism(s) by isn't studied in any great depth, it would appear that this effect is partially through a enrichment of the cell population into the more radiosensitive G2 phase of the cell cycle. Finally, the author's assess NEK1 expression in a relatively small (<100) cervical cancer patient cohort and find that higher levels of NEK1 correlate with poor prognosis, response to therapy and patient survival rates.
Overall, this paper is technically sound and the conclusions drawn from the results present are appropriate. Give the author's previous mechanistic study of NEK1 (Mol Cell, 2016), it would have been nice to see a more detailed mechanistic study of the effects of fractionation IR doses in the NEK1 proficient and deficient cells. It would also have been good to assess NEK1 clinicopathological associations in a larger patient cohort as well as an independent validation cohort.
My only main concern is that given the minimal amount of H2AX, cell cycle and caspase 3 data presented, the paragraph on line 272 is too strongly worded with regards to mechanistic insight, and should be changed accordingly.
Author Response
We would like to thank the reviewer for his/her kind evaluation of our manuscript and the helpful comment. Indeed, it was not the major purpose of our study to perform detailed mechanistic study to unravel the effects of fractionation IR doses in the Nek1 functionality, but to display a clinically applicable approach to increase radiation response mediated by Nek1 knockdown. In line with that, we totally agree with the reviewer’s objection that the paragraph is too strongly worded. Accordingly, in a revised version of our manuscript, we now more carefully state as follows (Page 9, lines 312-317): “Radiation sensitization by Nek1 KD is considered multifactorial involving multiple pathways. Thus, we next asked whether a Nek1 KD in the 6 h fractionated regime impacts on the DDR and apoptosis induction at 24 h after treatment. We observed a slight but significant (P < 0.01) increase in the number of residual (24 h) gH2AX foci in Nek1 KD compared with control cells (Figure 3A and 3B) together with a slightly increased fraction of apoptotic cell death assayed by a Caspase 3/7 activity assay (Figure 3C)”.